# Overexpression of *MdARD4* Accelerates Fruit Ripening and Increases Cold Hardiness in Tomato

**DOI:** 10.3390/ijms21176182

**Published:** 2020-08-27

**Authors:** Tianli Guo, Xiuzhi Zhang, Yuxing Li, Chenlu Liu, Na Wang, Qi Jiang, Junyao Wu, Fengwang Ma, Changhai Liu

**Affiliations:** State Key Laboratory of Crop Stress Biology for Arid Areas/Shaanxi Key Laboratory of Apple, College of Horticulture, Northwest A&F University, Yangling 712100, China; guo1003tianli@nwafu.edu.cn (T.G.); zxz0124@nwafu.edu.cn (X.Z.); lyx0707@nwafu.edu.cn (Y.L.); 2018050274lcl@nwafu.edu.cn (C.L.); wangna1993@nwsuaf.edu.cn (N.W.); jiangqi456123@sina.com (Q.J.); wjy97629@nwafu.edu.cn (J.W.)

**Keywords:** apple, *ARD4*, fruit ripening, cold tolerance, tomato

## Abstract

Ethylene plays an important role in stress adaptation and fruit ripening. Acireductone dioxygenase (ARD) is pivotal for ethylene biosynthesis. However, the response of ARD to fruit ripening or cold stress is still unclear. In this study, we identified three members of *Malus* ARD family, and expression profile analysis revealed that the transcript level of *MdARD4* was induced during apple fruit ripening and after apple plants were being treated with cold stress. To investigate its function in cold tolerance and fruit ripening, *MdARD4* was ectopically expressed in *Solanum lycopersicum* cultivar ‘Micro-Tom’, which has been considered as an excellent model plant for the study of fruit ripening. At the cellular level, the MdARD protein expressed throughout *Nicotiana benthamiana* epidermal cells. Overexpression of *MdARD4* in tomato demonstrated that *MdARD4* regulates the ethylene and carotenoid signaling pathway, increases ethylene and carotenoid concentrations, and accelerates fruit ripening. Furthermore, *MdARD4* increased the antioxidative ability and cold hardiness in tomato. To conclude, *MdARD4* may potentially be used in apple breeding to accelerate fruit ripening and increase cold hardiness.

## 1. Introduction

Ripening of fleshy fruits is a complex process that involves biochemical and metabolic changes. Ethylene, the natural plant hormone, regulates various physiological and developmental events in plants and is important for the ripening of climacteric fruits [1,2]. Ethylene and carotenoid concentrations are relevant to this stud related to fruit ripening. Tomato (*Lycopersicon esculentum* Mill.) has been considered as an excellent model plant for the study of fruit ripening [3]. Several studies have investigated the molecular basis of fruit ripening. Naeem et al. identified that *SmCOP1* (CONSTITUTIVE PHOTOMORPHOGENIC) regulates ethylene biosynthesis and affects fruit ripening in tomato [4]. Studies have also shown that ethylene triggers fruit ripening via the ethylene signaling pathway [5,6,7,8].

Plants are challenged by both biotic and abiotic stresses, including pathogen infection, salt stress, drought, and cold stress that limit their growth and development. These stresses pose significant threats to crop production worldwide [9,10,11]. As plants are sessile, they cannot escape from stress. Therefore, they utilize complex physiological and genetic regulatory networks to react to the continuously changing environmental conditions [9,10,12,13,14,15,16]. Apple (*Malus domestica* Borkh.) is one of the most important climacteric fruit crops worldwide. Extreme temperature, especially cold stress is one of the adverse environmental conditions that limit apple cultivation, growth, and development [17,18]. Low temperature influences the geographical distribution of apple and negatively impacts the productivity and quality of the fruit.

Ethylene, the fruit ripening hormone, also plays a key role in plant growth and development and regulates the plant response to stress. Different abiotic stress factors such as flooding, drought, high salinity, low temperature, and mechanical damage induce ethylene change in plants [19]. Researchers have investigated the association between enhanced ethylene levels and cold and freezing tolerance [20,21,22,23,24,25,26]. Ethylene is necessary in the normal response to cold stress in plants [27]. Overexpression of tomato ERF (ethylene-responsive factor) gene *TERF2/LeERF2* significantly activated stress-related genes and increased ethylene levels under the cold condition and thus enhanced cold tolerance [28]. Ethylene response factors such as VaERF080 and VaERF087 have been validated as cold-responsive factors. Transgenic *Arabidopsis* overexpression of VaERF080 and VaERF087 showed enhanced cold tolerance via lower malonyldialdehyde (MDA) content and higher superoxide dismutase (SOD), peroxidase (POD), and catalase (CAT) activities [29].

Similar findings have been reported in bermudagrass; CdERF1 improved cold tolerance by regulating stress-associated gene expression levels, relieving reactive oxygen species (ROS) damage, and improving the antioxidant system [30]. These earlier findings suggest that cold tolerance of a plant is particularly dependent on ethylene.

Different factors regulate ethylene syntheses. In plants, the Met cycle is required to sustain high rates of ethylene synthesis [31]. Methionine (Met) cycle exists in prokaryotes and eukaryotes. Acireductone dioxygenase (ARD) catalyzes the production of 2-keto-4-methylthiobuty-rate (KMTB) from acireductone and dioxygen and that affects ethylene synthesis [32]. ARD plays an important role in plant growth and development and stress response by participating in the Met cycle. It has been reported that OsARD4 improves root architecture in rice by promoting development of secondary roots [33]. TaARD may be involved in ethylene synthesis and ethylene signaling in response to biotic and abiotic stresses [34]. However, unlike the role of ethylene in fruit ripening and stress resistance, the function of ARDs, locates upstream of the ethylene biosynthetic pathway, in plants is not clear. Therefore, this study investigates the expression of *MdARD* in apple under the cold condition and during fruit ripening. To better understand the role of *ARD4* in regulating ethylene synthesis, fruit ripening, and cold hardiness, we ectopically expressed *MdARD4* in *Solanum lycopersicum* cultivar ‘Micro-Tom’, and then compared the differences of wild-type (WT) and transgenic plants in growth development and low temperature tolerance. These findings will provide valuable insights into *ARD4* regulation of cold hardiness and fruit ripening.

## 2. Results

### 2.1. Identification of MdARD Genes and Phylogenetic Analysis and Location Determination of MdARD Proteins

To identify the members of the ARD family in apple, previously reported ARD proteins of *Arabidopsis* were used as TBLASTN query to search the apple genome (version 1.0). Three proteins were identified using SMART and Pfam tools. To evaluate the evolutionary relationship between ARD proteins in apple, *Arabidopsis*, rice, pear, strawberry, potato, and grape, a phylogenetic tree was constructed based on the full length ARD protein sequences from these species. ARD of apple showed a close evolutionary relationship with *Arabidopsis* (Figure 1A). The ARD family proteins of apple were named based on their homologues in *Arabidopsis*. We identified and cloned three *Malus ARD* genes, namely *MdARD1*, *MdARD2*, and *MdARD4* with GenBank accession numbers MH481533, MH481534, and MH481535, respectively.

The MdARD proteins were expressed driven by a 35S promoter. The subcellular localization of GFP and MdARD-GFP proteins was examined by confocal microscopy. As shown in Figure 1B, the 35S-driven GFP protein showed expression throughout cell, and MdARD protein after fusion with GFP also showed expression throughout the cell.

### 2.2. Expression Pattern of MdARD Genes under Cold Condition and during Fruit Ripening in the Apple

To evaluate the role of *MdARD* genes under cold stress, their expression levels in the leaves of a ‘Golden delicious’ apple under cold stress were determined. *MdARD1* and *MdARD4* were significantly induced by cold while *MdARD2* did not change significantly under cold condition (Figure 2A). To evaluate the role of *MdARD* genes in fruit ripening, their expression levels in the fruit of the ‘Honey crisp’ apple during fruit ripening were determined. At 90 and 120 days after bloom, *MdARD* genes were significantly induced; *MdARD1* and *MdARD4* were increased during the fruit development process (Figure 2B). *MdARD4* showed higher expression under cold stress and during fruit ripening. Therefore, we chose *MdARD4* for further study.

### 2.3. Overexpression of MdARD4 Promotes Fruit Ripening in Tomato

Ethylene plays a key role in plant growth and development. ARD catalyzes the step of acireductone reacts with dioxygen to produce 2-keto-4-methylthiobuty-rate (KMTB) in the Met cycle and that affects the synthesis of ethylene. What about the function of MdARD4 in the fruit ripening process? We ectopically expressed *MdARD4* in *Solanum lycopersicum* cultivar ‘Micro-Tom’, and found that fruits of *MdARD4* transgenic tomato plants ripened faster than those of WT plants grown for the same duration under the same conditions (Figure 3A). Transgenic fruits differed from WT fruits in color. This change in color occurred at breaker stage; transgenic fruits turned yellow while WT remained green (Figure 3B). These results indicated that the ripening period was longer for WT plants than that for the transgenic plants (Figure 3C).

### 2.4. Overexpression of MdARD4 Improves Ethylene Concentration in Tomato

At the maturation stages corresponding to green and red fruit stages of tomato, *ARD4* relative expression level and ethylene release were higher in transgenic tomato fruits than those in WT fruits (Figure 4A,B). The expression levels of ethylene biosynthetic genes (*ACS2*, *ACO3*, *ACO1*, and *RIN*) and ethylene-responsive genes (*E4*, *E8*, and *ERF1*) were more also higher in *MdARD4* transgenic tomato fruits than those in WT fruits (Figure 4C,D).

### 2.5. Overexpression of MdARD4 Improves Carotenoid Concentration in Tomato

The color of transgenic tomato fruits differed from that of WT fruits at the same stage of maturation. Carotenoid concentration was higher in transgenic tomato fruits compared to that in WT fruits (Figure 5A). The expression of carotenoid biosynthetic genes (*PSY1*, *PDS*, and *ZDS*) was higher in the fruits of transgenic tomato (Figure 5B).

### 2.6. MdARD4-Overexpressing Tomato Lines Behave Better under the Cold Condition

To determine the role of *MdARD4* in plant cold resistance, we further compared the tomato plants grown under normal (25 °C) temperature and low (4 °C) temperature conditions. After 3 days of treatment, transgenic plants performed better than the WT plants (Figure 6A). Electrolyte leakage and malonyldialdehyde (MDA) concentration of leaves were measured to assess membrane damage under stress conditions. Under normal temperature, the relative electrolyte leakage and MDA concentration did not differ between the transgenic lines and the WT. However, they were significantly lower in the transgenic plants compared with the WT under cold condition (Figure 6B). The survival rates of the transgenic plant lines under low temperature conditions were considerably higher than that of the WT plants (Figure 6C). These results suggested that tomato plants overexpressing *MdARD4* sustained less damage and were cold tolerant.

### 2.7. Overexpression of MdARD4 Activates the Antioxidant System in Tomato under Cold Conditions

Stress-induced accumulation of highly toxic reactive oxygen species (ROS) can lead to oxidative stress, which damages various cell components. We monitored H_2_O_2_ levels of transgenic and WT plants leaves under both normal temperature and low temperature conditions. As shown in Figure 7, there were no visible differences in any parameter for any genotype grown under normal temperature conditions. However, after 5 days of 4 °C low temperature, transgenic plants showed significantly lesser H_2_O_2_ concentration compared with the WT plants (Figure 7A). There were no obvious differences in the antioxidant enzyme (SOD, CAT, and POD) involved in scavenging H_2_O_2_ activities in any genotype grown under normal temperature conditions, while the activities in transgenic plants were obviously more than that in the WT (Figure 7B–D). These results suggested that overexpressing *MdARD4* activates the antioxidant system under cold conditions.

## 3. Discussion

Ethylene plays a key role in plant growth, development, and stress resistance. The methionine (Met) cycle is required to sustain high rates of ethylene synthesis [31]. ARD catalyzes the step where acireductone reacts with dioxygen to produce 2-keto-4-methylthiobuty-rate (KMTB) and that affects the synthesis of ethylene [32]. The functions of ethylene in fruit ripening and stress and the underlying mechanisms have been reported in model plants. However, ARD’s roles in fruit ripening and abiotic stresses, such as cold stress, have been unexplored. In this study, we identified *MdARD* of *Malus domestica*. The ectopic expression of *MdARD4* in transgenic tomato suggested that *MdARD4* accelerates fruit ripening and increases cold hardiness.

Ethylene, which regulates various physiological and developmental events, is important for the overall fruit ripening process [1,2]. Ethylene biosynthesis is catalyzed by two key biosynthetic enzymes ACO and ACS [35,36,37,38]. In tomato, *SlACO1* and *SlACO3* were significantly induced during fruit ripening [6]. RNA interference of *SlACS2* in tomato fruits repressed ethylene production and fruit ripening [39]. Additionally, RIPENING INHIBITOR (RIN) and ethylene, via ethylene response factors (ERFs), are required for the complete expression of ripening genes such as *E4* and *E8*, the classic ethylene-induced genes [8]. ERFs regulate ethylene-dependent transcription and ethylene-inducible gene expression that play an important role in ripening [5,7,38,40,41,42,43,44]. Suppression of *E4* produced low-ethylene-producing phenotypes and increase in *E8* level induced ethylene production [45]. These results indicated that ethylene triggers fruit ripening via the ethylene signaling pathway. ARD locates upstream of the ethylene biosynthetic pathway. However, the function of *MdARD4* in fruit ripening is unknown and whether it is involved in ethylene biosynthesis and signaling pathway is unclear. It has been shown that overexpressing *OsARD1* enhanced the endogenous ethylene production in rice by upregulating ACC synthase genes including *OsACS2*, *OsACS4*, *OsACS5*, and *OsACS6* in *OsARD1*-overexpressed plants [46]. Similarly, this study, we found that overexpression of *MdARD4* in tomato promoted ripening by activating the expression of four ethylene biosynthetic genes *ACS2*, *ACO3*, *ACO1*, and *RIN*. Furthermore, the expression of ethylene-responsive genes *E4*, *E8*, and *ERF1* was more in transgenic fruits than in WT fruits compared at the same stage of maturity (Figure 4). Thus, overexpression of *MdARD4* induced the expression of genes involved in ethylene biosynthesis and response, improved ethylene concentration and shortened fruit maturation time. Further, higher concentration of carotenoids and enhanced expression of carotenoid biosynthetic genes were detected in transgenic fruits when compared with WT plants at the same stage of maturity (Figure 5). Previous studies have shown that overexpression of apple RNA binding protein MhYTP2, which interacts with MhARD4, accelerated fruit ripening in tomato [47]. This finding indicated that *MdARD4* might regulate fruit ripening by improving ethylene and carotenoid concentrations.

Cold, a common abiotic stress, influences the geographical distribution and negatively impacts productivity and quality of many important crops, such as an apple. Cold stress could reduce cell turgor and destabilize the plasma membrane of plants via ROS burst and oxidative damage, which result in cell damage and even plant death [48,49]. Electrolyte leakage and MDA concentration are typically used to evaluate the peroxidation of plasma membrane. In this study, the electrolyte leakage and MDA concentration were significantly lower in *MdARD4* transgenic tomato plants under cold stress (Figure 6). The antioxidative defense system of higher plants is one of the primary physiological response mechanisms under cold stress [50]. CAT, POD, and SOD, which have the ability to eliminate reactive oxygen species and protect the membrane, are essential indicators used to evaluate plant redox status. High CAT, POD, and SOD activities suggested high antioxidative ability and high cold resistance in *Arabidopsis* [29]. Numerous studies have shown that cold hardiness in various plants is affected by the levels of these antioxidant enzymes [51,52,53]. In this study, transgenic plants showed significantly decreased H_2_O_2_ concentration than WT plants after 5 days of 4 °C low temperature, and CAT, POD, and SOD activities were significantly higher in *MdARD4* transgenic tomato plants under cold conditions. Decreased H_2_O_2_ concentration is attributed to higher POD and CAT abilities. These findings indicated that *MdARD4* could increase the antioxidative ability, decrease the membrane lipid peroxidation, and improve the plasma membrane stability in plants in response to cold stress. On the one hand, overexpression of *MdARD4* improved ethylene concentration and shortened fruit maturation time, which may reduce the damage from cold stress due to the shorted growth stages. It may be an adaptation mechanism of the different biotic and abiotic stresses [54].

In conclusion, we identified three acireductone dioxygenase genes involved in both fruit ripening and cold hardiness: *MdARD1*, *MdARD2*, and *MdARD4*. *MdARD4* showed higher expression under cold stress and during fruit ripening. Overexpression of *MdARD4* accelerated fruit ripening in tomato via the regulation of ethylene and carotenoid signaling pathway and increase in ethylene and carotenoid concentrations. Furthermore, *MdARD4* transgenic tomato plants demonstrated better growth and cold tolerance compared with the WT plants partially due to the increase in antioxidative ability. This study provides new insights into the regulatory mechanism of ARD in fruit ripening and the cold stress response and proposes *MdARD4* as an ideal target candidate for breeding to regulate fruit ripening and cold tolerance in plants.

## 4. Material and Methods

### 4.1. Gene Cloning and Plasmid Construction

Leaves were collected from healthy ‘Golden delicious’ (*Malus domestica*) apple trees at the Horticultural Experimental Station of Northwest A & F University, Yangling (34°20′ N, 108°24′ E), China. These leaves were immediately frozen in liquid nitrogen and stored at −80 °C.

The open reading frames of *MdARD4* were obtained through expressed sequence tags (EST) assembly. RNA was extracted from the leaves using the CTAB method [55]. First-strand cDNA was synthesized using RevertAid™ First Strand cDNA Synthesis Kit (Fermentas, Burlington, ON, Canada) according to the manufacturer’s protocol. Primers for cloning were designed using Primer Premier 6.0 (PREMIER Biosoft Int., San Francisco, CA, USA). The sequences of primers used for cDNA cloning are listed in Appendix A. The amplified *MdARD4* product was inserted into the pCambia2300 plant expression vector under the control of the cauliflower mosaic virus (CaMV) 35S promoter. The plasmid was transformed into *Agrobacterium tumefaciens* EHA105 by the heat shock method [56] and was confirmed by sequencing.

### 4.2. Plant Materials and Treatments

Fruits of the ‘Honey crisp’ apple (*M. domestica*) were sampled for gene expression analysis at 0, 15, 30, 47, 61, 75, 92, 104, and 120 days after pollination between 3:00 PM and 4:00 PM. On each collection day, six apples per replicate were harvested from three trees, with a total of five replicates. The fruits were immediately weighed, cut into small pieces after removing the core and frozen on-site in liquid nitrogen. All frozen samples were stored at −80 °C.

The ‘Golden delicious’ apple grafted on the rootstock of *Malus hupehensis* and grown in pots at the Horticultural Experimental Station of Northwest A & F University, Yangling, China were used for expression analysis of *MdARD* under cold stress. To induce cold stress, the potted plants were placed at 4 °C and a 16 h photoperiod in the growth chamber. Mature leaves were sampled at 0, 2, 4, 8, and 12 h. Three biological replicates were collected for each treatment with five seedlings combined as one replicate. All collected tissues were immediately frozen in liquid nitrogen and stored at −80 °C.

### 4.3. Sequence Retrieval and Phylogenetic Tree Construction

The following databases were used to retrieve and compare the putative ARDs from the apple and *Arabidopsis*: Genome Database for Rosaceae (GDR; https://www.rosaceae.org/species/malus/malus_x_domestica) and *Arabidopsis* Information Resource (TAIR) database (http://www.Arabidopsis.org). *Arabidopsis* ARD sequences were used as queries to perform TBLASTN (Basic Local Alignment Search Tool; http://blast.ncbi.nlm.nih.gov) against the apple genome. We searched for ARD homologous protein sequences in other species using BLASTP. A phylogenetic tree was constructed using MEGA 5 software (http://www.megasoftware.net/) [57].

### 4.4. Subcellular Localization Analysis

*MdARD4* was cloned into the pGWB405 vector with a C-terminal green fluorescence protein (GFP). The fused vector and the empty vector were transformed into the *Agrobacterium tumefaciens* strain EHA105. Further, 5-week-old leaves of tobacco (*Nicotiana tabacum*) plants were transformed with these vectors. The infiltrated tobacco plants were grown for an additional 3 days under a 16 h:8 h, light:dark photoperiod at 21 °C in a growth chamber. The GFP signal in tobacco leaves was then detected by the Nikon A1R/A1 confocal microscope (Nikon, Tokyo, Japan).

### 4.5. Production of Transgenic Tomato Plants

The ORF of *MdARD4* was inserted into the pCambia2300 vector to construct the *MdARD4* overexpression vector. This construct was introduced into *Agrobacterium tumefaciens* and used to transform ‘Micro-Tom’ tomato at Wuhan Double helix Biology Science and Technology Co., Ltd. (http://www.bioslx.xom/; http://www.biogo.net/com/whslxgs/). Putative transgenic tomato plants were selected on the MS medium containing 50 mg·L^−1^ kanamycin. Further, 3:1 segregating lines were selected, and T3 homozygous transgenic lines were further confirmed by quantitative real-time PCR.

### 4.6. Wild-Type (WT) and Transgenic Plant Treatment

To analyze fruit ripening, fruits were sampled at the green fruit stage (G) and red fruit stage (R). The concentration of carotenoids and the expression levels of ethylene biosynthetic genes (*ACS2*, *ACO3*, *ACO1*, and *RIN*), ethylene-responsive genes (*E4*, *E8*, and *ERF1*), carotenoid biosynthetic genes (*PSY1*, *PDS*, and *ZDS*), and the internal control gene (*CAC*) were determined in the fruits at different developmental stages of wild type (WT) and transgenic (overexpressing *MdARD4*) tomato plants.

To further test the response of tomato plants to chilling, seeds of WT and transgenic tomato plants were sown in 250 cm^3^ plastic pots in a growth chamber under controlled conditions of 50% relative humidity, 28 °C and a long-day photoperiod (16 h:8 h, light:dark). The seedlings were watered regularly and supplied with half-strength Hoagland’s nutrient solution (pH 6.0) twice a week for 40 days to maintain growth. Seedlings of similar size were selected and divided into two groups. One group was placed in a growth chamber at 25 °C and the other group was placed in a growth chamber at 4 °C. After 5 days of treatment under the same photoperiod, H_2_O_2_ concentration, electrolyte leakage, MDA, and SOD, CAT, and POD activities were determined in the samples collected. The survival rates were determined as the number of visibly green plants 3 days after being under the normal conditions.

### 4.7. Ethylene Measurement

Ethylene production was measured according to the method by Han et al. with some modifications [58]. Five fruits were enclosed in a 3.6 L airtight vessel for 1 h at 25 °C, and 1 mL gas sample was collected with a syringe. This gas sample was injected into GC-14A flame ionization detector gas chromatograph (Shimadzu, Kyoto, Japan) to determine ethylene concentration. The ethylene production rate was expressed as µL·kg^−1^·h^−1^. Three biological replicates were maintained.

### 4.8. Analysis of MDA, Electrolyte Leakage and H_2_O_2_ Concentrations and Antioxidant Enzyme Activities

The concentration of MDA in the leaf extracts was determined by the thiobarbituric acid (TBA) method [59]. Electrolyte leakage was determined from leaves as described by [60,61] with an electrical conductivity meter (DSS-307; SPSIC, Shanghai, China). The level of H_2_O_2_ was measured as described by [62]. Leaf samples (0.1 g) were ground in a chilled mortar with 1% (*w*/*v*) polyvinylpolypyrrolidone and homogenized with 1.2 mL of 50 mM potassium phosphate buffer (pH 7.8) containing 1 mM EDTA-Na2 and 0.3% Triton X-100. Each homogenate was centrifuged at 13,000× *g* for 10 min at 4 °C. The supernatant was used to determine the activities of SOD, CAT, and POD following the established protocols [63,64,65,66,67].

### 4.9. RNA Extraction and Gene Expression Analysis

Total RNA from snap-frozen tissues was extracted by the CTAB method [55]. The total RNA was reverse transcribed into cDNA using PrimeScript ^®^ RT reagent Kit with gDNA Eraser (Perfect Real Time, Takara). Real-time quantitative PCR (Q-RT-PCR) was conducted on a LightCycler^®^ (Roche Ltd., Basel, Switzerland) 96 real-time PCR detection system (Roche) using the ChamQ SYBR qPCR Master Mixture (Vazyme biotech co., Ltd., Nanjing, China). Apple *ELONGATION FACTOR1α* (DQ341381)/*MdActin* (XM_008344381) and tomato *CAC* (NM_001324017)*/SlActin* (NM_001308447) were used as the internal control genes to compensate variation in cDNA concentrations and PCR efficiency between samples. The relative quantity of target gene transcript was determined by the 2^−ΔΔCt^ method [68,69]. The internal control genes were amplified in each sample and used for normalization. The results are presented in arbitrary units, as the ratio of the target gene expression to the expression of the reference gene in the indicated group was calculated as 1. All experiments had three biological replicates. Primers used for Q-RT-PCR are listed in Appendix A.

### 4.10. Statistical Analysis

Data were subjected to one-way analysis of variance (ANOVA), and mean differences were assessed by a Tukey test (*p* < 0.05).

## Figures and Tables

**Figure 1 ijms-21-06182-f001:**
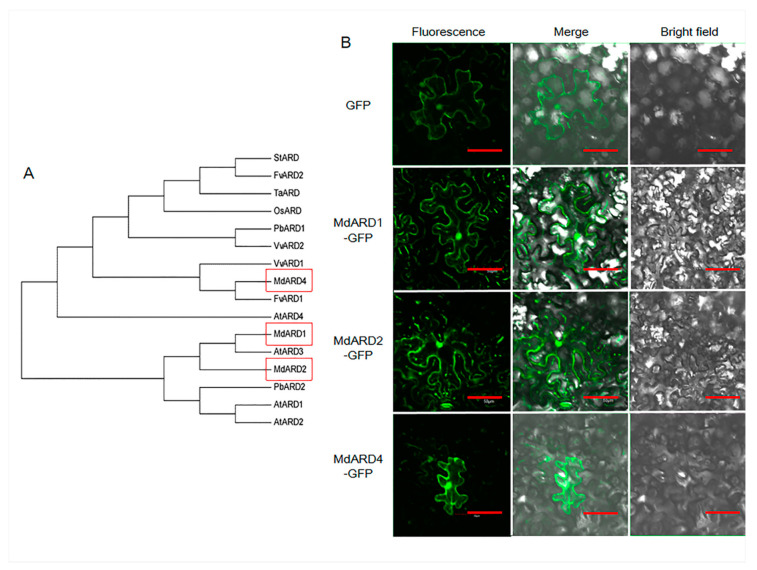
(**A**) Phylogenetic analysis among plant acireductone dioxygenase (ARD) proteins and (**B**) subcellular localization of GFP (green fluorescent protein) and MdARD-GFP. GFP or MdARD-GFP fusion protein were transiently expressed in *Nicotiana benthamiana* epidermal cells and visualized by confocal microscopy. The fluorescent green signal in the dark field or merged field shows the localization of GFP or MdARD-GFP fusion protein. Bars, 50 μm. Md, *Malus domestica*; *MdARD1* (MH481533), *MdARD2* (MH481534), *MdARD4* (MH481535). Pb, *Pyrus bretschneideri*; *PbARD1* (XP_009343958), *PbARD2* (XP_009375983). Fv, *Fragaria vesca*; *FvARD1* (XP_004296576), *FvARD2* (XP_004296575). Vv, *Vitis vinifera*; *VvARD1* (XP_010649963), *VvARD2* (XP_010649962). Os, *Oryza sativa*; *OsARD* (AAX55895). Ta, *Triticum aestivum; TaARD* (ACQ99820). St, *Solanum tuberosum;* StARD (NP_001274845). At, *Arabidopsis thaliana*; *AtARD1* (NP_567443), *AtARD2* (NP_001190730), *AtARD3* (NP_180208), *AtARD4* (NP_568630).

**Figure 2 ijms-21-06182-f002:**
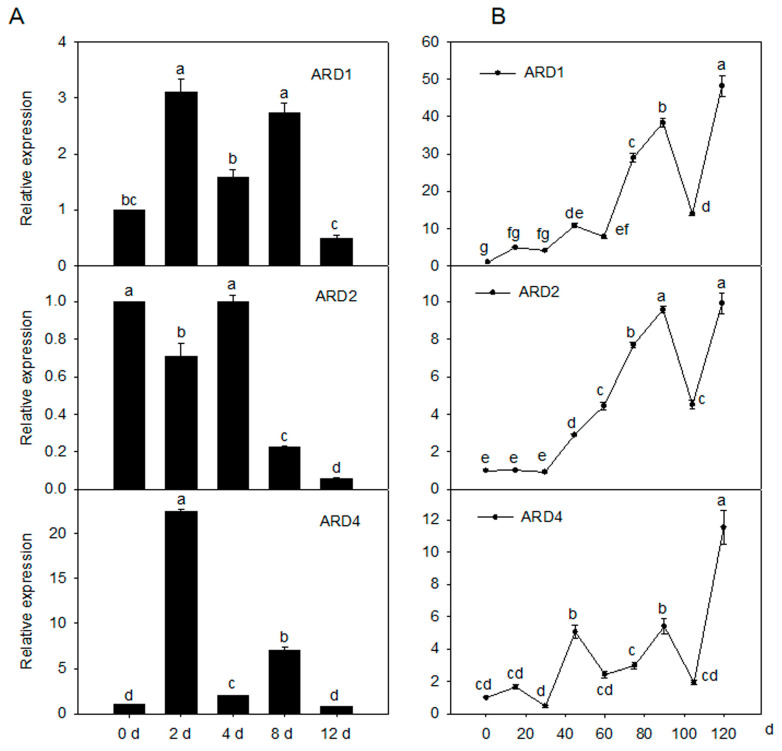
The expression pattern of *MdARDs.* (**A**) Change in the relative transcript level of *MdARDs* in response to the cold treatment, the expression level of *MdARD1*, *MdARD2*, or *MdARD4* in response to cold on Day 0 was set as 1 and (**B**) trends in relative expression of *MdARDs* at different developmental stages of the apple fruit. Samples from 0 days after bloom (DAB) were set to 1. Expression levels were calculated relative to that of *ELONGATION FACTOR1α* (*EF-1α*). Bars represent the mean value ± SE (*n* ≥ 3). Different letters indicate significant differences of *MdARDs* expression level on the different day of treatments, according to one-way ANOVA Tukey’s multiple range tests (*p* < 0.05).

**Figure 3 ijms-21-06182-f003:**
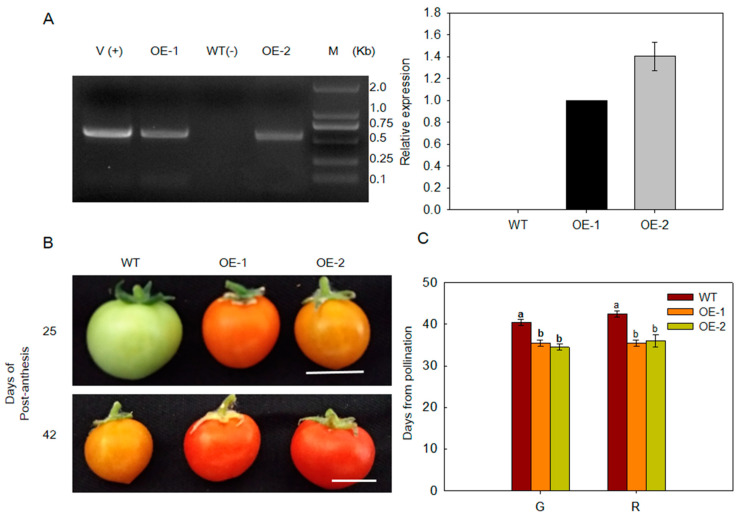
(**A**) PCR confirmation for transgenic tomato plants. **Left** panel: PCR with DNA; lanes: M, molecular marker DL2000; V(+), positive vector containing pCambia2300-*MdARD4* plasmid; WT(−), non-transformed wild type; OE-1 and OE-2, *MdARD4* transgenic tomato lines; **Right** panel: Quantitative RT-PCR analysis of *MdARD4* expression in leaves of WT and *MdARD4* transgenic lines OE-1 and OE-2, the expression level of MdARD4 in line OE-1 was set as 1; (**B**) phenotype comparisons between WT and *MdARD4* transgenic fruits. Bar, 2 cm; and (**C**) days from pollination to green (G) and red (R) stages. Data are means of three replicates with SD. Different letters indicate significant differences between WT and *MdARD4* transgenic plants on the same day of different treatments, according to one-way ANOVA Tukey’s multiple range tests (*p* < 0.05).

**Figure 4 ijms-21-06182-f004:**
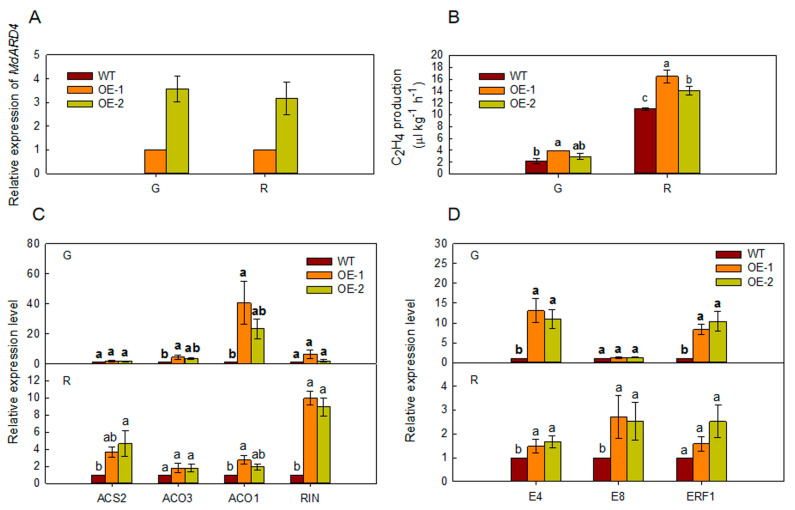
(**A**) Relative expression level of *ARD4* from WT and *MdARD4* transgenic tomato fruits, the expression level of MdARD4 in line OE-1 was set as 1 and (**B**) ethylene release level of *ARD4* from WT and *MdARD4* transgenic tomato fruits. Expression of ethylene biosynthesis genes (**C**) and ethylene-responsive genes (**D**) in fruits from WT and *MdARD4* transgenic tomato plants. Expression levels were calculated relative to that of *CAC*, the expression level of each gene in line WT was set as 1. Bars represent the mean value ± SE (*n* ≥ 3). Data are means of three replicates with SD. Different letters indicate significant differences between WT and transgenic plants on the same day of different treatments, according to one-way ANOVA Tukey’s multiple range tests (*p* < 0.05).

**Figure 5 ijms-21-06182-f005:**
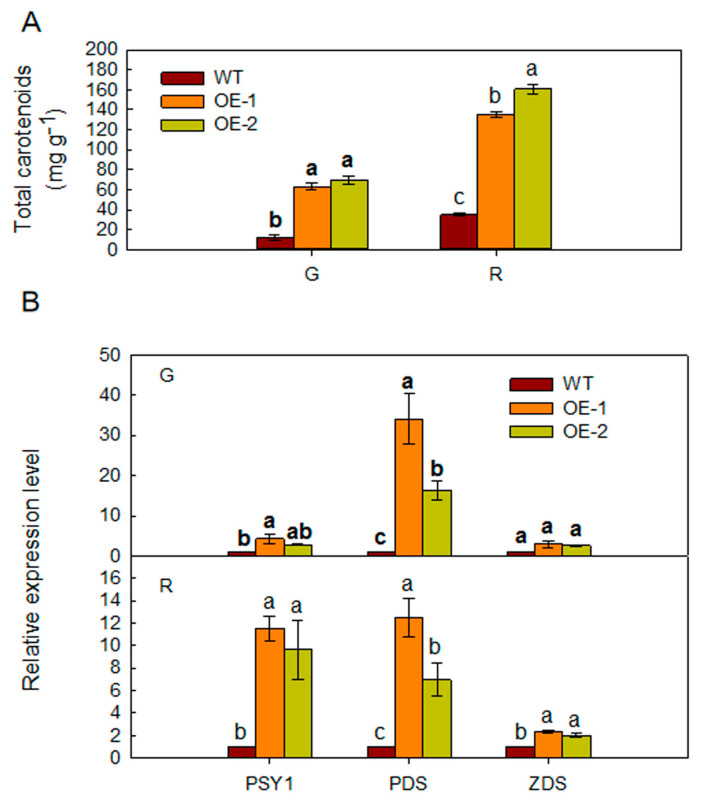
(**A**) The carotenoid concentrations in fruits from WT and *MdARD4* transgenic tomato plants and (**B**) expression of carotenoid biosynthesis genes in fruits from WT and *MdARD4* transgenic tomato plants, the expression level of each gene in line WT was set as 1. Data are means of three replicates with SD. Different letters indicate significant differences between WT and transgenic plants on the same day of different treatments, according to one-way ANOVA Tukey’s multiple range tests (*p* < 0.05).

**Figure 6 ijms-21-06182-f006:**
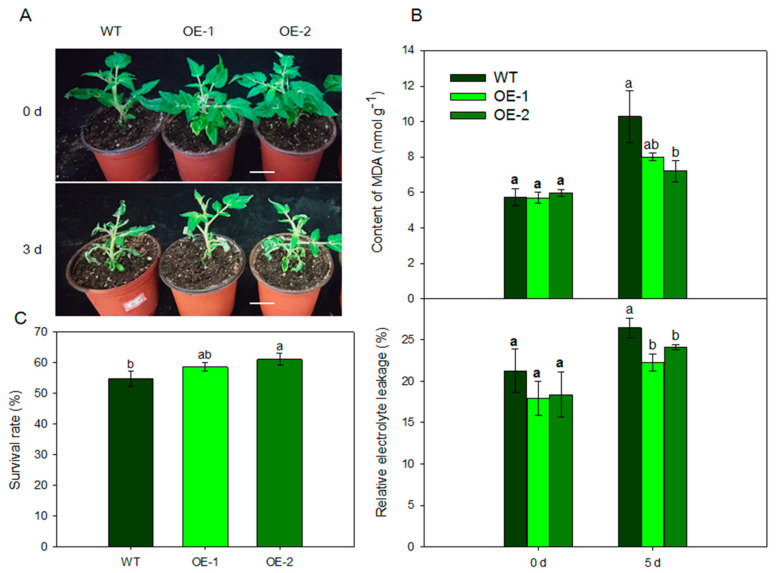
Effects of *MdARD4* overexpression in tomato on the degree of plant damage under cold conditions. (**A**) Phenotype comparisons between WT and *MdARD4* transgenic plants. Bar, 5 cm; (**B**) relative electrolyte leakage and MDA concentrations of WT and *MdARD4*-overexpression plants; and (**C**) survival rates of WT and *MdARD4*-overexpression plants. Data are means of three replicates with SD. Different letters indicate significant differences between WT and transgenic plants on the same day of different treatments, according to one-way ANOVA Tukey’s multiple range tests (*p* < 0.05).

**Figure 7 ijms-21-06182-f007:**
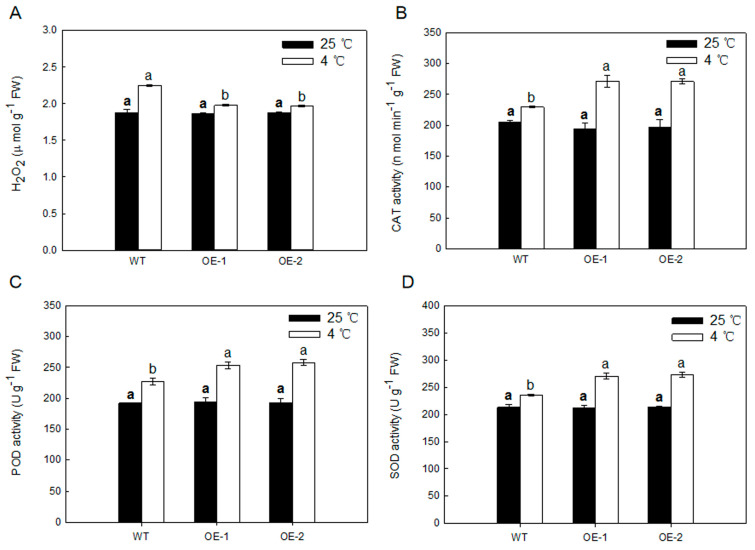
(**A**) Results of H_2_O_2_ concentrations in leaves from WT and *MdARD4* transgenic plants under both normal temperature and low temperature conditions and (**B**–**D**) antioxidative enzyme activities in leaves from WT and *MdARD4* transgenic plants under both normal temperature and low temperature conditions: CAT (**B**), POD (**C**), and SOD (**D**). Data are means of three replicates with SD. Different letters indicate significant differences between WT and transgenic plants on the same day of different treatments, according to one-way ANOVA Tukey’s multiple range tests (*p* < 0.05).

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
