# Peer review of "Overexpression of *MdARD4* Accelerates Fruit Ripening and Increases Cold Hardiness in Tomato"

_ijms, 2020, doi:10.3390/ijms21176182_

Round 1

Reviewer 1 Report

The manuscript entitled” Overexpression of MdARD4 Accelerates Fruit Ripening and Increases Cold Hardiness in Tomatothoroughly reviewed by me; I found that it has been written in very good language crisp and information compiled is up to date. The message of this manuscript is clearly delivered and concluded that MdARD4 regulates ethylene and carotenoid signaling pathway, increases ethylene and carotenoid concentrations, and accelerates fruit ripening. Furthermore, MdARD4 increased the antioxidative ability and cold hardiness in tomato. To conclude, MdARD4 can be potentially used in apple breeding to accelerate fruit ripening and increase cold hardiness. In this manuscript authors has explained each and everything with array of good experiments.  Discussion is bit weak and  have chances for improvement , Work is good and worth publishable, So after correction  and minor revision, it will be good information for the audience, and also this information can be used in future for solving queries related to role MdARD4 gene and its homologs in different plant systems.

Also I have few queries which are as following

Abstract

Line 3 Ethylene plays important roles in stress adaptation and fruit ripening. Acireductone

Line 14 dioxygenase (ARD) plays an important role in ethylene biosynthesis.Introduction: plays important role has been repeatedly came in sentence, so correct it.

In this study, we identify three members of Malus ARD  family after gene-cloning: it should be by gene cloning

Introduction

Line 32 Ripening of fleshy fruits is a complex a complex process that involves biochemical and metabolic….Correct it.

Line 35 concentration are relevant to this stud related to fruit ripening. Tomato (Lycopersicon esculentum Mill.).. should be 35 concentrations are relevant to this study.

Line 44 Therefore, they utilize complex physiological and genetic regulatory networks to react to the.. correct it.

Line 52 salinity, low temperature, and mechanical damage induce ethylene change inplants….fused words needs space

Line 53 have investigated the association between enhanced ethylene levelsand cold and freezing tolerance… fused words needs space

Line 73 the function of ARDs, located upstream of the ethylene biosynthetic pathway, in plants are not clear…it should be is not clear.

Discussion

Line 212 In this study, we found that ARD, located upstream of the ethylene biosynthetic pathway, activated
 Line 213 the expression of four ethylene biosynthetic genes ACS2, ACO3, ACO1, and RIN. ….is there are only these genes which are influenced more or few more have been reported in literature.

Discussion is bit weak there are chances to improve further.

Conclusion

Conclusion is not good, Authors rewrite crisp and clear conclusion so that it has good message for the audience.   

References

There are lots of error in references like scientific name is not italics, spacing issue somewhere formatting is also missing and somewhere full name of journals have been written, so please check the references to journal style while revising the manuscript. I have highlighted in PDF

Author Response

Point 1:

Abstract

Line 3 Ethylene plays important roles in stress adaptation and fruit ripening. Acireductone

Line 14 dioxygenase (ARD) plays an important role in ethylene biosynthesis.Introduction: plays important role has been repeatedly came in sentence, so correct it.

In this study, we identify three members of Malus ARD  family after gene-cloning: it should be by gene cloning

Introduction

Line 32 Ripening of fleshy fruits is a complex a complex process that involves biochemical and metabolic….Correct it.

Line 35 concentration are relevant to this stud related to fruit ripening. Tomato (Lycopersicon esculentum Mill.).. should be 35 concentrations are relevant to this study.

Line 44 Therefore, they utilize complex physiological and genetic regulatory networks to react to the.. correct it.

Line 52 salinity, low temperature, and mechanical damage induce ethylene change inplants….fused words needs space

Line 53 have investigated the association between enhanced ethylene levelsand cold and freezing tolerance… fused words needs space

Line 73 the function of ARDs, located upstream of the ethylene biosynthetic pathway, in plants are not clear…it should be is not clear.

Response 1: Firstly, thanks for your carefully reading and comments. We are sorry for the  grammar and space issues in the original manuscript. These have been carefully corrected and the new edits are highlighted in the manuscript.

Point 2:

Discussion

Line 212 In this study, we found that ARD, located upstream of the ethylene biosynthetic pathway, activated
Line 213 the expression of four ethylene biosynthetic genes ACS2, ACO3, ACO1, and RIN. ….is there are only these genes which are influenced more or few more have been reported in literature.

Discussion is bit weak there are chances to improve further.

Response 2: Thanks for your constructive suggestions. We have  literatures in accordance with your valuable suggestions  (Page10; Line 227-230).

It has been shown that overexpressing OsARD1 enhanced the endogenous ethylene production in rice by upregulating ACC synthase genes including OsACS2, OsACS4, OsACS5, and OsACS6 in OsARD1-overexpressed plants [46].

And we have added some information in the Discussion part (Page10; Line 258-261).

On the one hand, overexpression of MdARD4 improved ethylene concentration and shortened fruit maturation time, which may reduce the damage from cold stress due to the shorted growth stages. It may be an adaptation mechanism of the different biotic and abiotic stresses (Tao et al., 2015).

Point 3:

Conclusion

Conclusion is not good, Authors rewrite crisp and clear conclusion so that it has good message for the audience.   

Response 3: Thanks for your comments. The conclusion has been carefully revised to make it crisp and clear. The new edits are highlighted in the revised manuscript. We hope the modified conclusion can bring good message to the readers.   

Point 4:

References

There are lots of error in references like scientific name is not italics, spacing issue somewhere formatting is also missing and somewhere full name of journals have been written, so please check the references to journal style while revising the manuscript. I have highlighted in PDF

Response 4: We are very sorry for the errors in references. The references have been carefully checked and revised. The new edits are highlighted in the revised manuscript.

Reviewer 2 Report

- Major Compulsory Revisions

Firstly, I want to indicate, that International Journal of Molecular Sciences (IJMS) is journal with high Impact Factor, therefore, authors should include new experimental data, should improve described data presented in manuscript (Ms) text and should improve discussion and conclusions.

1) Please, present the Ms without comments. It is difficult to read Ms with comments.

2) Authors should improve all used figures and legend for figures, e.g.:

a) In Fig. 1a: Authors should mark three used apple acireductone dioxygenase (ARD) genes (MdARD1, MdARD2, MdARD4). Also, in the 1a legend authors should present the GeneBank accession numbers of all used ARD proteins.

b) In Fig. 1b: “Bars = 50μm.” I did not find bars in Fig. 1.

c) Authors should increase the quality of the Fig. 1b. Explain, I see 3 columns of figures, how do they differ? Also, I see MdARD1, MdARD2, MdARD4 proteins (as I understand fusion protein with GFP), but I did not see GFP protein alone.

d) Fig. 2: Authors should include statistical treatment in Fig. 2. Also, where are the values of gene expression on day 140 (Fig. 2b).

e) Fig. 3a left: I see the signal in the negative. The authors should redo this result.

f) Fig. 3a right: what sample was set for 1?

g) Fig. 3b left: bar size?

h) Fig. 3b right: authors should explain “GT” and “RT” abbreviations in Fig. 3 legend.

I) Fig. 4, 5, and 7: authors should improve statistical treatment in these figures.

J) Fig. 6a: Authors should increase figure quality and include bar.

3) In this version of the Ms it is difficult to find the complete sequence of the studied genes. Authors should include GeneBank accession number of the MdARD4 gene.

4) In Ms “Expression levels were calculated relative to that of ELONGATION FACTOR1α (EF-1α; DQ341381)”. Usually, in real time PCR at least 2 internal controls are used, therefore authors should include second internal control.

5) Authors should include data about effect of gene overexpression on growth and yield of transgenic tomatoes

6) Authors should include data about viability, healthy green seedlings, or survival rate of the non transgenic and MdARD4-transgenic tomatoes after cold treatment like in Fig. 6b (Franz et al., Molecular Plant, 2011) or in  Fig. 2b (Dubrovina et al., Plant Cell Tissue and Organ Culture, 2016).

- Minor:

7) Line 493: “63Li,” correct to “Li,”.

8) Line 332: “The relative quantity of target gene transcript was determined by 2−ΔΔCT method.” Include reference for this method.

Author Response

Point 1:

1)Please, present the Ms without comments. It is difficult to read Ms with comments.

Response 1: We are very sorry for the presented comments in the original manuscript. They have been removed.

Point 2:

2)Authors should improve all used figures and legend for figures, e.g.:

  1. a) In Fig. 1a: Authors should mark three used apple acireductone dioxygenase (ARD) genes (MdARD1, MdARD2, MdARD4). Also, in the 1a legend authors should present the GeneBank accession numbers of all used ARD proteins.
  2. b) In Fig. 1b: “Bars = 50μ” I did not find bars in Fig. 1.
  3. c) Authors should increase the quality of the Fig. 1b. Explain, I see 3 columns of figures, how do they differ? Also, I see MdARD1, MdARD2, MdARD4 proteins (as I understand fusion protein with GFP), but I did not see GFP protein alone.
  4. d) Fig. 2: Authors should include statistical treatment in Fig. 2. Also, where are the values of gene expression on day 140 (Fig. 2b).
  5. e) Fig. 3a left: I see the signal in the negative. The authors should redo this result.
  6. f) Fig. 3a right: what sample was set for 1?
  7. g) Fig. 3b left: bar size?
  8. h) Fig. 3b right: authors should explain “GT” and “RT” abbreviations in Fig. 3 legend.
  9. I) Fig. 4, 5, and 7: authors should improve statistical treatment in these figures.
  10. J) Fig. 6a: Authors should increase figure quality and include bar.

Response 2:  We are very sorry for the bad resolution of figures. The quality of figures has been improved. The new edits are highlighted in the revised manuscript.

  1. a) In Fig. 1a: We have marked the three apple acireductone dioxygenase (ARD) genes and the GeneBank accession numbers of all used ARD proteins are provided in figure 1a legend.
  2. b) We have remarked the barsin Fig. 1.
  3. c)  We have improved the quality of Fig. 1b and we hope that the modified figures are satisfied. With the current subcellular localization results, we found that all three ARDs appeared in the entire cell. We did not find any differences among these three proteins based on our observation. Additionally, the subcellular localization of GFP protein alone has been added.
  4. d) Fig. 2: The statistical treatment has been added in Fig. 2. There were no values of gene expression on day 140 (Fig. 2b). We are very sorry for making confusion about gene expression on day 140 in Fig. 2b. The Fig. 2b has been revised.
  5. e) Fig. 3a left: The signal in the negative belongs to the wide-type, which acts as negative control.
  6. f) Fig. 3a right: The transgenic overexpression of MdARD4 OE-1was set for 1.
  7. g) Fig. 3b left: Bar = 2cm, which is represented as white lines in the figure and bar size is described in figure legend.
  8. h) Fig. 3b right: “GT” and “RT”have beenedited to “G” and “R”and their abbreviations are described in Fig. 3 legend.
  9. I) Fig. 4, 5, and 7: The statistical treatment in these figures has been improved.
  10. J) Fig. 6a: We are very sorry for the low quality pictures. We have increased the figure quality and included bar.

Point 3:

3) In this version of the Ms it is difficult to find the complete sequence of the studied genes. Authors should include GeneBank accession number of the MdARD4 gene.

Response 3: Thanks for your constructive suggestions. We have included GeneBank accession numbers of the three MdARD genes (Page2;Line 89-91).

Point 4:

4) In Ms “Expression levels were calculated relative to that of ELONGATION FACTOR1α (EF-1α; DQ341381)”. Usually, in real time PCR at least 2 internal controls are used, therefore authors should include second internal control.

Response 4: We also used “MdActin” /“SlActin” as gene expression internal controls for apple and tomato, respectively. We got similar consistent gene expression trends. We presented new gene expression data using actin as internal control. Figures corresponding to Fig. 2, Fig. 3A right, Fig. 4A, C, D and Fig. 5B are shown below.

Point 5:

5) Authors should include data about effect of gene overexpression on growth and yield of transgenic tomatoes

Response 5: Thanks. It is really a wonderful and constructive suggestion. However, we did not find significant differences in growth and yield between the transgenic and wild type plants. We only focus on fruit ripening and cold hardiness.

Point 6:

6) Authors should include data about viability, healthy green seedlings, or survival rate of the non transgenic and MdARD4-transgenic tomatoes after cold treatment like in Fig. 6b (Franz et al., Molecular Plant, 2011) or in  Fig. 2b (Dubrovina et al., Plant Cell Tissue and Organ Culture, 2016).
Response 6: Thanks for your constructive suggestions. We have added data about viability, healthy green seedlings of the non transgenic and MdARD4-transgenic tomatoes after cold treatment in Fig. 6c.

Round 2

Reviewer 2 Report

Tianli Guo et al. “Overexpression of MdARD4 accelerates fruit ripening and increases cold hardiness in tomato”, International Journal of Molecular Sciences, ijms-882619-R1.

This review paper describes expression analysis of three acireductone dioxygenase (ARD) genes in apple Malus domestica and overexpression of the one MdARD4 gene from apple in tomato Solanum lycopersicum. Therefore, the topic of this manuscript (Ms) is relevant for International Journal of Molecular Sciences (IJMS). Authors improved the Ms text compare with previously version. However, I have several critical remarks to the Ms. Ms needs Major Revisions.

Author Response

Dear reviewer,

Thank you very much for reviewing our manuscript. We appreciate your comments which not only helped us a lot in revising this ms to fit for publication but also inspired us a new direction for future study. We have carefully revised the manuscript according to your comments and a point-to-point response is listed below. If you have further comments, please do not hesitate to let us know. Thanks again.

Sincerely,

Fengwang Ma & Changhai Liu

Point 1:

1) “Fig. 3a left: I see the signal in the negative. The authors should redo this result.” Author responds: e) Fig. 3a left: The signal in the negative belongs to the wide-type, which acts as a negative control.”

- Once again, I see some signal in this negative sample. Authors should sequence this PCR product or present figure 3a without this PCR product in WT(-).

Response 1: Thank you. We are sorry that we misunderstood your meaning in the first round. We did the PCR again and a new figure 3A without this PCR product in WT(-) is presented now.

Point 2:

2) “Author response 4: We also used “MdActin” /“SlActin” as gene expression

internal controls for apple and tomato, respectively. We got similar consistent gene expression trends. We presented new gene expression data using actin as internal control. Figures corresponding to Fig. 2, Fig. 3A right, Fig. 4A, C, D and Fig. 5B are shown below.”

- Line 353-355 – I see “ELONGATION FACTOR1α and tomato CAC were used

as the internal control genes.”. Usually, in real time PCR at least 2 internal controls are used, therefore authors should include second internal control. Authors should present 2 internal controls (GeneBank accession numbers and primers) for apple and tomato.

Response 2: Thanks for your suggestion. We have included GeneBank accession numbers for "MdActin" and "SlActin" gene ( on Page12; Line 357-358). And the corresponding primers are shown in the Appendix A.

Point 3:

3) “Authors should include data about effect of gene overexpression on growth and yield of transgenic tomatoes. Author response 5: Thanks. It is really a wonderful and constructive suggestion. However, we did not find significant differences in growth and yield between the transgenic and wild type plants. We only focus on fruit ripening and cold hardiness.”

- Where in the Ms text did the authors present this data. Is it better to present this data as a table?

Response 3: We are very sorry that we confused you. Since MdARD4 could clearly accelerate tomato fruit ripening when being overexpressed, we absolutely agree it would be better to indicate the effect of MdARD4 on plant growth and yield. Regretfully, during our experiment, we did not pay attention to the growth and yield. Therefore, we do not know whether MdARD4 overexpression would affect growth and yield or not. We do appreciate your suggestion, we will pay attention to the growth and yield in our future study.

Point 4:

4) “Authors should include data about viability, healthy green seedlings, or survival rate of the non transgenic and MdARD4-transgenic tomatoes after cold treatment like in Fig. 6b (Franz et al., Molecular Plant, 2011) or in Fig. 2b (Dubrovina et al., Plant Cell Tissue and Organ Culture, 2016). Author response 6: Thanks for your constructive suggestions. We have added data about viability, healthy green seedlings of the non transgenic and MdARD4-transgenic tomatoes after cold treatment in Fig. 6c.”

- Fig. 6c - what are the dots before the numbers and survival rate in %? Did I understand correctly that the transformation only increased survival rate by 0.5-1 %? Are these values really statistically different?

Response 4: Thank you for your comment. Yes, readers may misunderstand the previous version of figure 6C, we have changed the title for Y axis to “suvival rate (%)” and modified the label numbers to “0, 10, 20, ...”. The MdARD4 transformation increased survival rate by 5-10 % which are statistically different compared with WT. Similarly, for Figure 6B (lower panel), we also changed the title for Y axis to “relative electrolyte leakage (%)” and modified the label numbers to “0, 5, 10, 15, ...”. Moreover, we also modified Y axis labels of figures 2A (middle panel), 3A (right panel) and figures 7A which do not have 0 before the dot.

Point 5:

5) Authors should verify statistical treatment in all presented figures (e.g. why no error bars in some probes in 3a right, 4a, 4b).

Response 5: Thanks for the comment. In our current study, all the gene expression data was determined by 2−ΔΔCT method. In Figure 3A (right panel) and 4A, the expression level of MdARD4 in line OE-1 was set as 1, so the error value was 0. In Fig. 4B, the SD value of ethylene release from MdARD4 transgenic tomato fruits (OE-1) is 0.01231, the error bar is so short which is overlapped with the column. For figure 2A, the expression level of MdARD1, MdARD2 or MdARD4 in response to cold on Day 0 was set as 1. For figure 2B, the expression level of MdARD1, MdARD2 or MdARD4 during different fruit development stages on Day 0 was set as 1. In figure 4C, 4D and 5B, the expression level of each gene in line WT was set as 1.

Round 3

Reviewer 2 Report

1) Line 355-356: “The relative quantity of target gene transcript was determined by 2−ΔΔCT method [68].” Reference 68 is Kim et al. 2009. But in this paper in Material and methods I did not find “real time PCR”, only “Semiquantitative RT–PCR”. Authors should explain used 2−ΔΔCT method.

2) Again, authors should verify statistical treatment in all presented figures (e.g. why  “c, ab, a” letters in Fig. 6c? May be “b, a, a”?).

Author Response

Dear reviewer,

Thanks a lot for your comments. We have carefully revised the manuscript according to your comments and our response is listed below. If you have further comments, please do not hesitate to let us know. Thanks again.

Sincerely,

Fengwang Ma & Changhai Liu

Point 1:

1) Line 355-356: “The relative quantity of target gene transcript was determined by 2−ΔΔCT method [68].” Reference 68 is Kim et al. 2009. But in this paper in Material and methods I did not find “real time PCR”, only “Semiquantitative RT–PCR”. Authors should explain used 2−ΔΔCT method.

Response 1: Thank you very much. Actually, the reference 68 used both “Semiquantitative RT–PCR” and “real time PCR” to determine gene expression levels, although the authors did not describe the method for “real time PCR” in Materials and Methods. For example, figure 4B (bottom panel) showed the real-time PCR result. The reason we cited this paper is that the authors normalized the level of RLuc mRNA in the mock-depleted extract as 100 which is the same as what we did in this study. Since Ref 68 did not mention 2−ΔΔCT method, we added another reference, number 69, which clearly explained the 2−ΔΔCT method (on Page12; Line 363-366).

Point 2:

2) Again, authors should verify statistical treatment in all presented figures (e.g. why  “c, ab, a” letters in Fig. 6c? May be “b, a, a”?).

Response 2: Thank you very much. We have carefully checked the statistics for all the figures and did find mistakes presented in 2 figures. The letters should be “b, ab, a” for figure 6C and “b, a, ab” for figure 4B (G stage). We are very sorry for these mistakes and thank you for your carefully examination.
